# Homisland-IO: Homogeneous Land Use/Land Cover over the Small Islands of the Indian Ocean

**Christophe Révillion [1,*], Artadji Attoumane [2] and Vincent Herbreteau [3]**

[1]  Université de La Réunion, UMR 228 Espace-Dev (IRD, UA, UG, UM, UR),
    40 Avenue de Soweto 97410 Saint-Pierre, La Réunion, France
[2]  Institut de Recherche pour le développement, UMR 228 Espace-Dev (IRD, UA, UG, UM, UR),
    40 Avenue de Soweto 97410 Saint-Pierre, La Réunion, France; attoumane.artadji@ird.fr
[3]  Institut de Recherche pour le développement, UMR 228 Espace-Dev (IRD, UA, UG, UM, UR),
    5 Preah Monivong Blvd (93), Phnom Penh, Cambodge; vincent.herbreteau@ird.fr
**\***  Correspondence: christophe.revillion@univ-reunion.fr; Tel: +262-62-966-952

**Abstract:** Many small islands are located in the southwestern Indian Ocean. These islands have their own environmental specificities and very fragmented landscapes. Generic land use products developed from low and medium resolution satellite images are not suitable for studying these small territories. This is why we have developed a land use / land cover product, called Homisland-IO, based on remote sensing processing on high spatial resolution satellite images acquired by SPOT 5 satellite between December 2012 and July 2014. This product has been produced using an object-based classification process. The overall accuracy of the product is 86%. Homisland-IO is freely accessible through a web portal and is thus available for future use.

**Keywords:** land cover; remote sensing; object-based image analysis; small island; Indian Ocean

---

## 1. Summary

Land use / land cover (LULC) data are key information to understand the relationships between humans and their environment. As such, these data are used in many research fields, for example, climate change [1,2], biodiversity assessment [3], health [4,5] or land use and land cover changes monitoring [6,7].

To carry out this work successfully, it is necessary to obtain the most appropriate data for the territory and topic studied. A global land cover, such as the MODIS global land cover with a spatial resolution of 500 m [8] or the Landsat TM global land cover with a spatial resolution of 30 m [9] may be sufficient to study large areas on a continental or national scale, but their spatial resolutions are not adapted for smaller territories.

In the southwestern Indian Ocean there are many islands less than 3000 sq. km (Figure 1): the Seychelles Archipelago (with 115 islands, including Mahé, the largest), the Comoros Archipelago (Grande Comore, Mohéli, Anjouan and Mayotte), the Scattered Islands (the Glorioso Islands, Juan de Nova, Bassas da India, Europa, Tromelin Island), the Mascarene Islands (Réunion, Mauritius, Rodrigues). These small island territories have very fragmented and diversified landscapes. Also, global land use products with spatial resolutions ranging from 4 km to 30 m and therefore with heterogeneous pixels whose spectral response is derived from various objects, are not or are only poorly adapted to study these areas [8–10]. In contrast, there are specific land use products that only concern some of these islands [11–13] or a part of them [14,15]. This is what led us to produce a very

high spatial resolution LULC product for the Indian Ocean's small islands (Comoros, Seychelles, Mascarenes, scattered islands), that we called Homisland-IO (HOMogeneous land use product on the small ISLANDs of the western Indian Ocean). We analysed SPOT 5 images, acquired between December 2012 and July 2014, using object-oriented remote sensing methods. With a spatial resolution of 2.5 m in panchromatic mode, these images are suitable for precise mapping of certain LULC classes, particularly for urban areas [16].

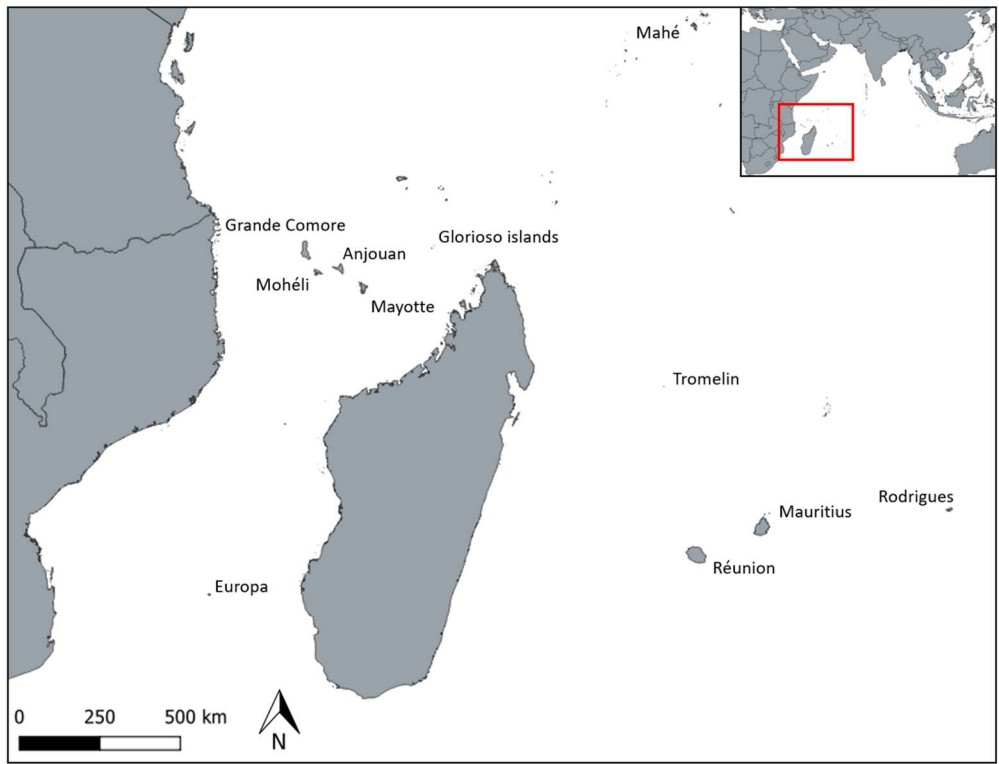

**Figure 1.** Location of the southwestern Indian Ocean small islands and the 11 islands produced.

## 2. Data Description

### 2.1. Homisland-IO LULC Maps

Homisland-IO offers LULC classification maps in vector format for 11 islands of the Indian Ocean: Anjouan, Europa, Glorioso Islands, Grande Comore, Mauritius, Mahé, Mayotte, Mohéli, Réunion Island, Rodrigues Island and Tromelin. The LULC typology is divided into 11 classes that partly correspond to the first level of classification established by Anderson *et al.* for Landsat images [17]. The definition of these classes is similar to those of Anderson et al. For example, urban classes are artificialized as are impermeable spaces, bare soils are areas of fine soil, sand or rocks [17]. Nevertheless, the typology has been adapted to better represent the LULC on these tropical islands. To be more precise, the "rangeland" class has been renamed to "shrub vegetation", the tundra class to "herbaceous vegetation" and the wetland class to "mangrove". Finally, the "perennial ice" class was not retained; it was considered that this type of land cover was not really interesting in tropical areas. The nomenclature was also adapted to the study area by dividing the agricultural class into two classes - a "sugar cane" class and an "other cropland" class - to better reflect the sugar cane class, which is present and even dominant on Réunion Island and Mauritius.

### 2.2. Homisland-IO Accuracy

A confusion matrix and a Kappa index describe the accuracy of each map. The Kappa values range between 0.77 and 0.93, which is satisfactory (Table 1). We brought together the scattered islands (Glorioso Islands, Tromelin and Europa) for this calculation due to their small size.

**Table 1.** Kappa index per island.

| Réunion | Mohéli | Anjouan | Eparses | Mauritius | Mahé | Mayotte | Grande Comore | Rodrigues |
|---------|--------|---------|---------|-----------|------|---------|---------------|-----------|
| 0.85 | 0.86 | 0.89 | 0.93 | 0.85 | 0.92 | 0.82 | 0.77 | 0.89 |

The overall confusion matrix also compiles the results of each island matrix and provides a high 0.86 Kappa index (Table 2).

**Table 2.** Overall confusion matrix (1, continuous urban; 2, Discontinuous urban; 3, Forest; 4, shrub vegetation; 5, herbaceous vegetation; 6, mangrove; 7, barren land; 8, open water area; 9, sugarcane; 10, pasture; 12, other cropland).

| Class Number | 1 | 2 | 3 | 4 | 5 | 6 | 7 | 8 | 9 | 10 | 12 | Sum |
|--------------|-----|-----|-----|-----|-----|-----|-----|-----|----|----|-----|------|
| 1 | 104 | 5 | 1 | 1 | | | 5 | | | | | 116 |
| 2 | 2 | 266 | 5 | 4 | 6 | 2 | 15 | | | | | 300 |
| 3 | | 2 | 389 | 29 | 1 | | 1 | | | 1 | 1 | 424 |
| 4 | | 2 | 41 | 267 | 32 | 1 | 2 | | | | | 345 |
| 5 | | 4 | 9 | 27 | 242 | | 16 | | | | | 298 |
| 6 | 2 | | 10 | 3 | | 230 | 5 | 5 | | | | 255 |
| 7 | | 7 | 3 | 1 | 17 | | 256 | 9 | | | | 293 |
| 8 | | | | | | 5 | 4 | 311 | | | | 320 |
| 9 | | | 1 | | | | | 1 | 63 | | 3 | 68 |
| 10 | | | 3 | | | | | | | 29 | 1 | 33 |
| 12 | | | 2 | 5 | 6 | 1 | 1 | | 5 | | 111 | 131 |
| | | | | | | | | | | | | |
| Sum | 108 | 286 | 464 | 337 | 304 | 239 | 305 | 326 | 68 | 30 | 116 | 2583 |
| **Kappa = 0.86** | | | | | | | | | | | | |

The classes with the lowest accuracy are the three vegetation classes, each of which are confused with each other. This is mainly due to the heterogeneity of plant formations on these islands.

### 2.3. Homisland-IO Availability

The eleven LULC maps can be viewed in the dedicated Homisland-IO web portal (http://homisland.seas-oi.org/). The user can quickly select each map, zoom in for details and see the nomenclature and metadata (Figure 2).

Moreover, these maps are available for download on both the Zenodo platform (https://zenodo.org/record/2585747) and the Homisland-IO web portal to allow an easy access to researchers and mathematicians using LULC in their work. They are freely accessible and reusable under a Creative Commons license (CC by 4.0). Here, we provide these maps in a shapefile format, so they can be used in a geographic information system for analysis or producing further maps. Each map is also accompanied by an XML file describing its metadata and corresponding to the INSPIRE directive schema, and also the confusion matrix and Kappa index.

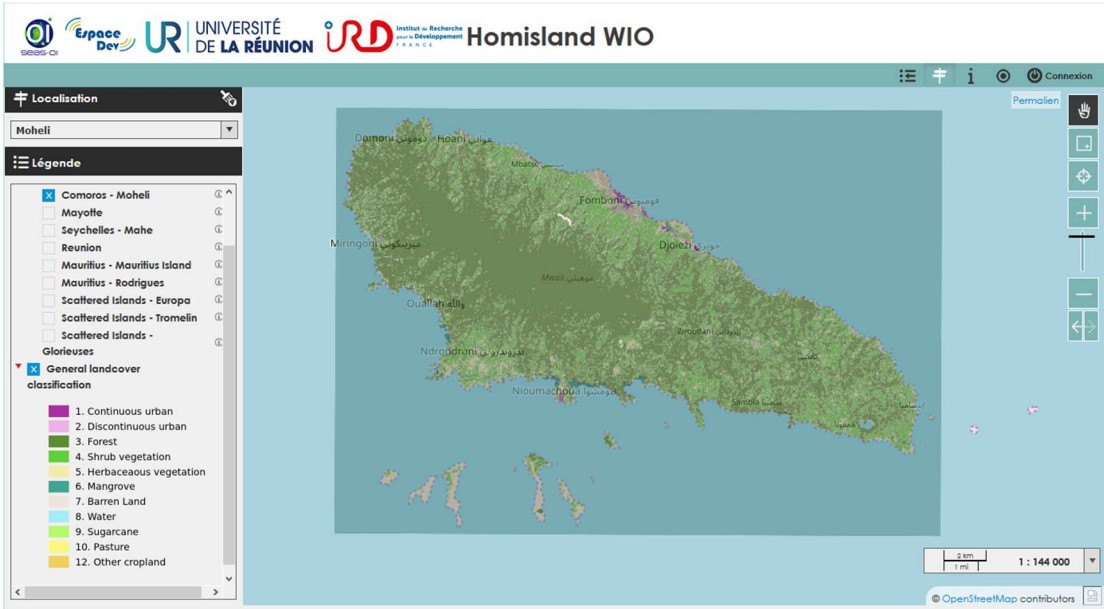

**Figure 2.** Example of a landcover on the lizmap web portal.

## 3. Methods

### 3.1. Data Used

The SEAS-OI (Survey of the environment Assisted by Satellite in the Indian Ocean) station at La Réunion provided nearly cloud-free SPOT 5 satellite images that it acquired between 2012 and 2014 (Table 3). These images are freely accessible to public institutions and researchers based in these islands through its catalogue (http://www.seas-oi.org/web/guest/catalogue#/main).

**Table 3.** Information on SPOT 5 satellite data used.

| Island | Acquisition Date | Cloud Cover (%) |
|---|---|---|
| Grande Comore | May 30th 2013 | 2% |
| Anjouan | August 13th 2014 | 3% |
| Mohéli | December 8th 2014 | 0% |
| Mayotte | July 16th 2013 | 5% |
| Réunion | May 21st 2014 | 5% |
| Europa | June 24th 2013 | 0% |
| Mauritius (two images) | May 17th 2014 | 0% |
| Rodrigues | May 18th 2014 | 0% |
| Tromelin | April 25th 2014 | 0% |
| Glorioso Islands | November 8th 2013 | 0% |
| Mahé | December 6th 2012 | 3% |

We used the multispectral (10 m) and panchromatic (2.5 m) pair to have a 2.5 m spatial resolution for all available bands. The images are of good quality with very few clouds, which is rare for optical images in intertropical areas. However, on the south of the Grande Comore image there is a problem on a pixel line; this problem comes from the satellite sensor and is found on the classification product. We integrated the Shuttle Radar Topography Mission (SRTM, downloaded from the SRTM Tile Grabber: http://dwtkns.com/srtm/), into the classification process to distinguish mangroves on the coast from shaded forest areas.

To improve the classification results, we used some geographic data from the free and participative OpenStreetMap (OSM: https://openstreetmap.org) database. We first verified and completed (when needed) the data to ensure their quality. This information has enabled us to improve our results for agricultural lands (sugar cane, vegetable gardening) and to more precisely

define the main roads. More precisely, the data integrated into the classification process correspond to the OSM keys and values: "landuse=farmland", "highway=primary road" and "highway=trunk road".

*3.2. Classification*

The general methodology and software used are described in Figure 3.

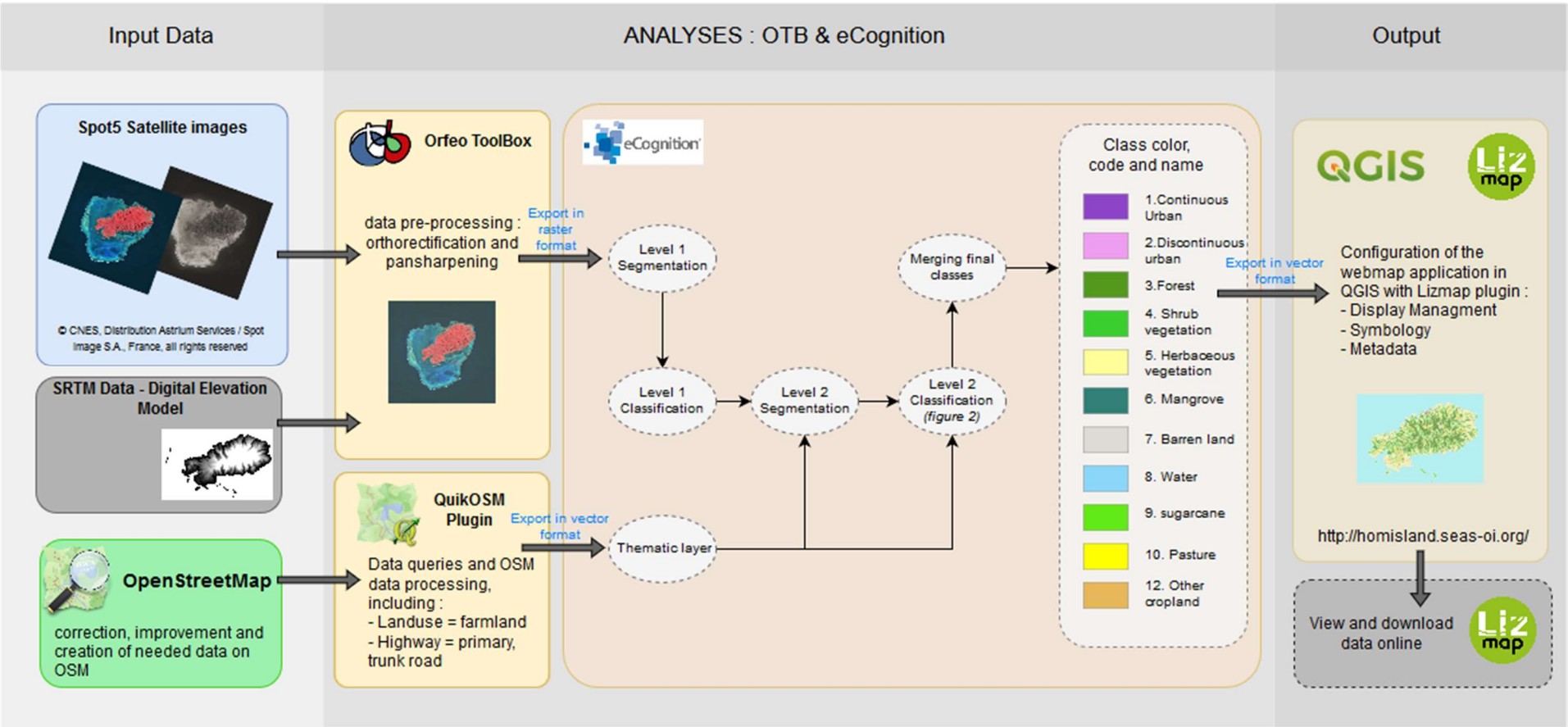

**Figure 3.** Processing chain to generate and diffuse the land use/land cover.

3.2.1. Preprossessing of Images.

We first corrected the geometry of these images to make them superimposable. We applied the same geometric correction to the thematic data to take into account the relief. We used the orthorectification application of the OrfeoToolBox 5.8.0 (OTB, https://www.orfeo-toolbox.org/) open source software for this preprocessing.

3.2.2. OBIA Classification Using Membership Functions

We chose to classify images using an Object-Based Image Analysis (OBIA) method. Unlike the traditional "per pixel" classification, which focuses mainly on spectral signature of surface states, the OBIA approach operates groups of contiguous pixels based on the spectral value, shape, length-to-width ratio, etc., of the objects to be classified [18]. This classification method is particularly suitable for high-resolution images [19]. The OBIA approach is divided into two main stages: segmentation and classification. Segmentation consists of cutting an image into group pixels according to certain homogeneity criteria. These are calculated according to several parameters such as color (spectral value) and shape. A scale criterion is then used to determine the maximum allowed heterogeneity [20]. Classification makes it possible to group objects by formulating a number of membership rules that can be combined, based on the observation and knowledge of the remote sensor. Each object is described by attributes related to its reflectance, texture, geometry and context (relationships with neighboring objects).

We used eCognition software 9.0.3 (http://www.ecognition.com/) to realize these segmentations and classifications.

Segmentation

Among the four types of segmentation algorithms available in eCognition, we chose the multi-resolution segmentation since it is the only one that allows the use of data from various sources and types as well as multi-scale analysis [18,21]. The segmentation parameters used are detailed in Table 4. These settings were generally the same for all images. They were empirically determined by visual analysis and integrated into the "multiresolution segmentation" algorithm of the eCognition software. The objective of these segmentations was to obtain very homogeneous objects at the spectral level.

**Table 4.** Segmentation parameters.

|                 | Level 1 | Level 2 |
| --------------- | ------- | ------- |
| Scale parameter | 40      | 10      |
| Shape           | 0.1     | 0.1     |
| Compactness     | 0.5     | 0.8     |

Classification

We performed a "hierarchical classification" using membership functions in eCognition at the two levels of the segmentation. At the first level, we separated open water areas (ocean, lakes) from land areas. We used the average near-infrared channel (mean NIR) as a criterion. At the second level, we classified land areas according to a class hierarchy by using several discrimination criteria (Figure 4).

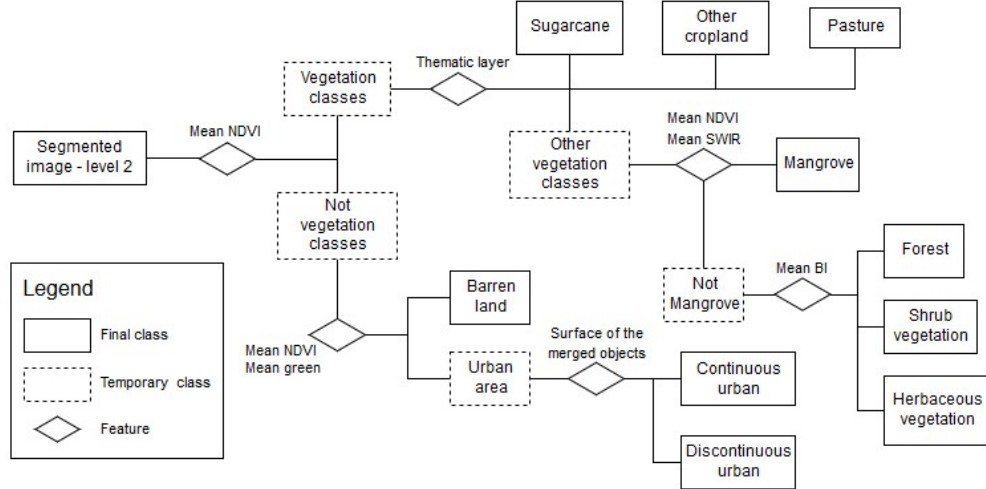

**Figure 4.** Class hierarchy of the classification at level 2. BI is the Brightness Index, NDVI the Normalized Difference Vegetation Index, SWIR the Short-Wave InfraRed.

The detection thresholds of the classes vary according to the study areas, with the histogram of radiometric values of the images being different from one image to another. These thresholds were defined manually with the help of the tools provided by the eCognition software. Nevertheless, the criteria used to separate the different classes are very similar for all classification processes (Figure 4).

3.2.3. Postprocessing

We realized a post-classification by photo-interpretation in order to correct some confusion between bare grounds and buildings, particularly on bare ground areas that have a strong overall reflectance (sand for example).

We made the LULC data consistent for dissemination using QGIS software. We merged contiguous objects corresponding to the same class to enhance the visualization and make easier future use in vector format.

We finally informed a metadata XML file following the INSPIRE directive schema, using the QSphere plugin for QGIS.

3.2.4. Accuracy Assessment

We conducted a ground truth survey in each island except in the scattered islands, which are difficult to access. For these islands, we used the very high spatial resolution images provided by Google Earth to obtain control points. The points were entered manually with a GPS; for practical reasons they were generally taken not far from the communication routes and sometimes along pedestrian paths when possible. We collected between 30 and 50 control points per class and per island to correctly assess the accuracy of each classification [22]. The choice of the size of the ground control sites is determined in part by the pixel size of SPOT5 images. For example, we did not take 3 isolated trees to make a ground control point for the forest class. We computed a confusion matrix for each island to compare observed to classified values by class (Table 1). We calculated Cohen's Kappa for each LULC map, which indicates the difference between the observed accuracy and what could be expected by chance. A map is considered accurate when this index is over 75% [23].

The confusion matrix and Kappa index of each map are attached to the dataset available for download.

*3.3. Discussion*

The accuracy indicates a good quality of the Homisland-IO maps. Nevertheless, there is still some room for improvement in several points. Agricultural classes have not yet been included in Comoros' LULC because the dominant crop is agroforestry. Much more field data and higher

resolution satellite data will be needed to better discriminate this type of coverage [24]. The agricultural classes on the Comoros are currently grouped into forest and shrub vegetation classes. Also, it would be interesting to differentiate between natural grasslands and pasture areas throughout the dataset in order to better reflect human activities. These two types of information could be acquired by photo-interpretation using OSM.

The OBIA classification process under the eCognition software with the integration of high quality exogenous data has resulted in the production of robust LULC data. These maps are representative of the small and fragmented territories of the small islands of the southwest Indian Ocean. These data can be key information to improve the understanding of human activities in these small territories. They are also important basic data for territories that do not have a national geographical database, such as the Union of the Comoros.

**Author Contributions:** Christophe Révillion led the conceptualization and the processing of the data; Christophe Révillion and Artadji Attoumane developed and validated the data; Vincent Herbreteau contributed to the methods and the discussion of the results; Vincent Herbreteau supervised the projects that funded this work; Christophe Révillion wrote the paper; Vincent Herbreteau and Artadji Attoumane reviewed and edited the manuscript.

**Funding:** Several projects that required these LULC maps have helped to fund the production of Homisland-IO:
- LeptOI project (FEDER POCT 31569);
- ISSE-Mayotte (French Ministry of Outremer, MOM 2012);
- IRD for the funding of a PhD thesis (ARTS) in the Comoros;
- TROI project (FEDER InterReg V).

**Acknowledgments:** The authors would like to thank:
- The SEAS-OI platform for the distribution of SPOT 5 images and Didier Bouche (Université de La Réunion) for his support in the development of the web portal;
- The 3Liz company (3liz.com) for its remarkable free and opensource webmapping tool, lizmap for QGIS and, together with Etienne Trimaille, for the very useful QuickOSM plugin for QGIS;
- OSM contributors and especially those at La Réunion who contributed to this mapping.

**Conflicts of Interest:** The authors declare no conflict of interest. The funders had no role in the design of the study; in the collection, analyses, or interpretation of data; in the writing of the manuscript, or in the decision to publish the results.

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
