# Peer review of "Homisland-IO: Homogeneous Land Use/Land Cover over the Small Islands of the Indian Ocean"

_data, 2019_

Round 1

Reviewer 1 Report

Line 79, missing “be”, between can and viewed,

Line 92, SEAS-OI is not explained,

Line 144, PIR is not explained,

Line 165, there is no description of the sample design

Author Response

Dear reviewer,

Thank you for your review. we have corrected your comments in the new version of the manuscript. We have taken into account the corrections requested:

1. Line 92 SEAS-OI means Survey of the environment Assisted by Satellite in the Indian Ocean;

2. Line 144 PIR means NIR (near infrared) in french;

3. Also we have detailed better the method set up for the ground truth.

Best Regards

Reviewer 2 Report

see enclosed file bellow.

Author Response

Dear reviewer,

Thank you for your review. we have corrected your comments in the new version of the manuscript. We have taken into account the corrections requested :

2. Line 139 the visualization problem has been fixed;

3. Line 144 PIR means NIR (near infrared) in french, so we corrected and spelled out;

4. Line 146 IB is the Brightness Index (BI) in french, we have corrected this in the figure and explained the acronym in the figure description;

5. At the beginning we had 12 land use classes but we removed the orchard class because it was poorly represented on our study areas and rather poorly classified. It was class 11, that's why the other cropland class remained class 12. We have removed class 11 on the website. We have noted that in a future version of the data it will be necessary to recode class 12 into 11;

6. Mangroves are detected mainly with the NDVI index and the SWIR channel. We have added this information in Figure 3;

7. The two urban classes were separated according to the surface area of the merged objects. This threshold has also been adapted according to the islands. We have added this information in Figure 3

Best regards

Reviewer 3 Report

Comments:

Line 144: Please spell out PIR.

Figure 3: What is IB?

Line 149-152: How thresholds are determined for each island. Is there any training data?

Line 165-166: Are control points randomly generated or manually selected? That might impact the accuracy assessment. Will the validation data be shared as well?

Land cover product:

The "continuous urban" often mixed with "barren" or "herbaceous vegetation". Is the urban here means impervious surface? I suggest authors list out definitions of classes.

There is a straight line across southern Grande Comoros, which seems like an artificial effect. But the manuscript suggested only one image was used.  So more interpretation is needed.

Author Response

Dear reviewer,

Thank you for your review. we have corrected your comments in the new version of the manuscript. We have taken into account the corrections requested :

1. Line 144 PIR means NIR (near infrared) in french, so we corrected and spelled out;

2. Figure 3 IB Is the Brightness Index (BI) in french, we have corrected this in the figure and explained the acronym in the figure description;

3. Line 149-152 the thresholds were defined manually with the help of the tools provided by the eCognition software. There is no training data.

4. Line 165-166 : The points were entered manually with a GPS, for practical reasons they were generally taken not far from the communication routes and sometimes along pedestrian paths when possible. We have added this information to the manuscript;

5. On the land cover product : Yes, Urban classes represent artificialized and impervious surfaces.

The definition of the classes is similar to the description of Anderson and al..

6. On the land cover product : The straight line in the south of the greater comoros is on the raw spot5 image. There was probably a problem during the acquisition. The input dataset is very good but it still has some small problems. We have added a short explanation in the section "data used";

Best regards